# Variant of Concern-Matched COVID-19 Convalescent Plasma Usage in Seronegative Hospitalized Patients

**DOI:** 10.3390/v14071443

**Published:** 2022-06-30

**Authors:** Massimo Franchini, Daniele Focosi, Elena Percivalle, Massimiliano Beccaria, Martina Garuti, Omar Arar, Antonietta Pecoriello, Fabio Spreafico, Graziana Greco, Stefano Bertacco, Marco Ghirardini, Tiziana Santini, Michele Schiavulli, Muzzica Stefania, Thaililja Gagliardo, Josè Camilla Sammartino, Alessandro Ferrari, Matteo Zani, Alessia Ballotari, Claudia Glingani, Fausto Baldanti

**Affiliations:** 1Department of Hematology and Transfusion Medicine, Carlo Poma Hospital, 46100 Mantua, Italy; matteo.zani@asst-mantova.it (M.Z.); alessia.ballotari@asst-mantova.it (A.B.); claudia.glingani@asst-mantova.it (C.G.); 2North-Western Tuscany Blood Bank, Pisa University Hospital, 56124 Pisa, Italy; daniele.focosi@gmail.com; 3Microbiology and Virology Unit, Fondazione IRCCS Policlinico San Matteo, 27100 Pavia, Italy; e.percivalle@smatteo.pv.it (E.P.); c.sammartino@smatteo.pv.it (J.C.S.); alessandro.ferrrari@smatteo.pv.it (A.F.); 4Intensive Care Respiratory Unit, Carlo Poma Hospital, 46100 Mantua, Italy; massimiliano.beccaria@asst-mantova.it (M.B.); martina.garuti@asst-mantova.it (M.G.); omar.arar@asst-mantova.it (O.A.); antonietta.pecoriello@asst-mantova.it (A.P.); fabio.spreafico@asst-mantova.it (F.S.); graziana.greco@asst-mantova.it (G.G.); stefano.bertacco@asst-mantova.it (S.B.); 5Department of Medicine, Carlo Poma Hospital, 46100 Mantua, Italy; marco.ghirardini@asst-mantova.it (M.G.); tiziana.santini@asst-mantova.it (T.S.); 6Regional Reference Center for Coagulation Disorders, Santobono-Pausilipon Children’s Hospital, AORN, 80129 Naples, Italy; mischiavulli@gmail.com; 7Pediatric Emergency and Short Stay Unit, Santobono-Pausilipon Children’s Hospital, AORN, 80129 Naples, Italy; s.muzzica@santobonopausilipon.it (M.S.); t.galiardo@santobonopausilipon.it (T.G.); 8Department of Clinical, Surgical, Diagnostic and Pediatric Sciences, University of Pavia, 27100 Pavia, Italy; fausto.baldanti@unipv.it

**Keywords:** COVID-19 convalescent plasma, SARS-CoV-2, variants of concern, efficacy

## Abstract

COVID-19 convalescent plasma (CCP) has been the only specific anti-viral therapy against SARS-CoV-2 available for more than one year. Following the negative results from most randomized controlled trials on its efficacy in COVID-19 hospitalized patients and the availability of anti-spike monoclonal antibodies (mAbs), the use of CCP has subsequently rapidly faded. However, the continuous appearance of new variants of concern (VOCs), most of which escape mAbs and vaccine-elicited neutralizing antibodies (nAbs), has renewed the interest towards CCP, at least in seronegative immunocompetent patients, and in immunocompromised patients not able to mount a protective immune response. We report here the experience of a single Italian hospital in collecting and transfusing CCP in immunocompromised patients hospitalized for severe COVID-19 between October 2021 and March 2022. During this 6-month period, we collected CCP from 32 vaccinated and convalescent regular blood donors, and infused high nAb-titer CCP units (titered against the specific VOC affecting the recipient) to 21 hospitalized patients with severe COVID-19, all of them seronegative at the time of CCP transfusion. Patients’ median age was 66 years (IQR 50–74 years) and approximately half of them (47.6%, 10/21) were immunocompromised. Two patients were rescued after previous failure of mAbs. No adverse reactions following CCP transfusion were recorded. A 28-day mortality rate of 14.3 percent (3/21) was reported, with age, advanced disease stage and late CCP transfusion associated with a worse outcome. This real-life experience also supports the use of CCP in seronegative hospitalized COVID-19 patients during the Delta and Omicron waves.

## 1. Introduction

COVID-19 convalescent plasma (CCP) has been the only antibody-based therapy available against SARS-CoV-2 infection for more than one year (January 2020–March 2021). Following the first positive experiences from non-randomized controlled trials (RCTs), its use rapidly spread around the world [1,2]. Further data collected from RCTs have, however, globally reported the lack of efficacy of CCP in seropositive patients hospitalized for severe COVID-19 [3], hence its use was discouraged by the World Health Organization (WHO) and many scientific societies [4,5,6] and has rapidly declined since the third pandemic wave. Contrary to this trend, based on the safety of CCP [7,8] and on the positive signals of efficacy in subgroup analyses from RCTs and non-RCTs, [9] the US Food and Drug Administration (FDA) approved the Emergency Use Authorization (EUA) for high-titer COVID-19 convalescent plasma to patients with immunosuppressive disease or receiving immunosuppressive treatment [10].

In Italy, the experience of CCP use has been overall positive. Besides the aforementioned early proof of concept trial [2], real-life data from Italian Expanded Access Program (EAP) trials documented the safety and efficacy of this passive immunotherapy [11,12]. Additionally, the experiences of selected categories of patients, such as onco-hematologic patients, were encouraging [13]. In addition, the Italian RCT TSUNAMI showed a trend towards CCP efficacy in the subgroup of hospitalized COVID-19 patients with milder disease [14,15]. In spite of this evidence, CCP was also abandoned in our country [6], as recommended by the Italian Society of Immunohematology and Transfusion Medicine (SIMTI). The concomitant difficulty in accessing anti-spike monoclonal antibodies (mAbs) left many fragile patients without an antibody-based therapy against SARS-CoV-2 variants of concern (VOCs) and was probably a contributor to the high rate of COVID-19-related deaths recorded in Italy [16]. In the City Hospital of Mantua, based on the previous positive experiences [2,11,17], clinicians continued to order CCP during Delta and Omicron pandemic waves, albeit to a lesser extent and in restricted categories of patients, such as seronegative elderly and immunocompromised patients. To meet the demands of the clinicians, we improved the quality of the CCP product by enhancing its potency (higher nAb titer) and specificity (against VOCs widespread in that geographical area at that time). In particular, we have collected CCP locally from vaccinated people recovered from COVID-19 as recently as possible. In addition, due to the widespread diffusion of SARS-CoV-2 infection in Italy at the end of 2021 and the beginning of 2022, we managed to collect CCP from voluntary regular blood donors, thus improving the overall safety of plasma [18]. The collection of CCP was therefore driven by transfusion requests, keeping stocks to the minimum necessary. We present here the results of collection and clinical use of CCP at the City Hospital of Mantua during the period October 2021–April 2022. 

## 2. Material and Methods

### 2.1. Donors’ Selection

CCP donors were selected among the 17,200 regular blood donors belonging to the network of the Italian Association of Voluntary Blood Donors (AVIS) within the province of Mantua. Healthy COVID-19 vaccinated donors aged 18–65 years and weighing more than 50 kg, with a history of positive SARS-CoV-2 PCR test (molecular identification of VOC) and who had recently recovered from COVID-19 (negative SARS-CoV-2 PCR test with nasopharyngeal swab), were recruited. To prevent transfusion-related acute lung injury (TRALI), previously transfused blood donors or female donors with a history of pregnancies were excluded from CCP donation, according to the Italian legislation [19]. All routine screening tests for blood donors, including ABO blood group typing, Rh phenotype, complete blood cell count, and screening for human immunodeficiency virus, hepatitis B and C, and syphilis were conducted according to Italian regulations and the indications of the Italian National Blood Center [19]. For each donor, the following parameters were recorded: sex, age, day of CCP donation, COVID-19 vaccine type (BNT162b2 Pfizer-BionTech, New York, NY, USA, mRNA-1273 Moderna, Cambridge, USA, ChAdOx1 AstraZeneca, Cambridge, UK, Ad26.COV2.S Janssen-Johnson & Johnson, Titusville, NJ, USA), number and date of vaccine doses, anti-SARS-CoV-2 nAb titer (titered against wild type, Delta, and Omicron VOCs), days between CCP donation and last vaccine dose and COVID-19 recovery, and VOC lineage of the recipient.

### 2.2. Patients’ Selection

All COVID-19 patients were hospitalized and enrolled in the frame of a CCP compassionate use program authorized by both the local ethical committee and the hospital health management. The study was registered at clinicaltrials.gov as NCT05157165. All patients had a confirmed diagnosis of SARS-CoV-2 infection (i.e., with a nasopharyngeal swab positive for SARS-CoV-2 by PCR) and at least one of the following criteria indicative of severe COVID-19: (1) radiologically confirmed pneumonia; (2) oxygen saturation (SpO2) ≤ 93% at rest and in room air; (3) partial pressure of oxygen (PaO2)/fraction of inspired oxygen (FiO2) ≤ 300 mmHg. Written informed consent was obtained before enrollment. Patients with proven hypersensitivity or allergic reaction to plasma, blood product, or immunoglobulins were excluded from CCP transfusion.

In addition to patients’ demographic and physical data (age, sex, and body mass index (BMI)), previous/concomitant therapies and comorbidities were also considered. In addition to these parameters, we recorded the date of hospital admission, the COVID-19 vaccine type and the number of doses administered, days since last COVID-19 vaccine and SARS-CoV-2 infection, VOC lineage, degree of COVID-19 severity, department of hospitalization (low intensity: infectious disease and internal medicine departments, where patients underwent medical therapy, including oxygen therapy at high flows, but not mechanical ventilation; intermediate–high intensity: respiratory intensive care unit, emergency medicine and intensive care unit where patients underwent non-invasive mechanical ventilation (NIMV) and invasive mechanical ventilation (IMV)), PaO2/FiO2 ratio before CCP transfusion, anti-SARS-CoV-2 IgG, number and anti-SARS-CoV-2 nAb titer of CCP units transfused, days between symptom onset and first CCP transfusion, days between last CCP transfusion and viral clearance (negative nasopharyngeal swab), and the outcome (mortality) 28 days after hospitalization.

### 2.3. COVID-19 Convalescent Plasma

CCP was collected through a plasmapheresis procedure using the AURORA cell separator (Fresenius Kabi, Verona, Italy), frozen, and stored in agreement with national CCP regulations [19,20]. A plasma volume of about 600 mL was collected during each procedure and immediately divided into two bags, each corresponding to a therapeutic CCP unit of 300 mL. We transfused collected CCP units with an anti-SARS-CoV-2 nAb titer of 1:160 or higher. The CCP transfusions were performed by medical and nursing staff. CCP recipients were transfused with one to three units of ABO type compatible CCP, according to the clinical response, over a period of 3–5 days. All the procedures were performed in agreement with the routine procedures of the Transfusion Service of the City Hospital of Mantua. All CCP donors provided their written consent after being thoroughly informed.

### 2.4. Laboratory Tests

VOC identification for both donors and recipients was performed using an in house primer set to identify variants showing mutations in the spike sequence as reported in Gaiarsa et al. [21]. Briefly, total RNA was extracted using the QIAamp Viral RNA Mini Kit according to the manufacturer’s instructions. The extracted RNA was subjected to a one-step RT-PCR using the SuperScript IV One-Step RT-PCR System (Thermo Fisher Scientific, Waltham, MA, USA).

Serologic tests (anti-SARS-CoV-2 IgG antibodies) were performed at the Laboratory Center of the City Hospital of Mantua, and viral neutralization tests (VNTs) were performed at the Molecular Virology Unit of the University Hospital of Pavia. Donors’ plasma samples were collected at the time of CCP donation.

Quantitative determination of SARS-CoV-2 anti-S1 and anti-S2 IgG antibodies was performed in patients using chemiluminescent immunoassay (CLIA) LIAISON^®^ SARS-CoV-2 S1/S2 IgG (DiaSorin, Saluggia, Italy), according to the manufacturer’s instructions. Results were given as arbitrary units (AU)/mL, and a cut-off of 15 AU/mL was considered for the definition of positive samples. Results ranging from 12 to 15 AU/mL were considered borderline or weak positive and IgG titers < 12 AU/mL were given as a negative result.

VNTs were run in a BSL-3 laboratory by experienced personnel. All variants used were isolated from nasopharyngeal swabs of infected symptomatic patients. To confirm the presence of variant-defining mutations, whole genome sequencing was performed and sequences were submitted to GISIAD [22]. After isolation, all variants were propagated in a VERO E6 25 cm^2^ cell culture flask (Corning Incorporated, Corning, NY, USA) to increase virus titer and prepare virus stock for VNTs; the titer of each variant was measured at the 50% tissue culture infectious dose (TCID50) in six replicas, determined by applying the Reed–Muench method [23].

VNTs were performed as previously described [24]. Briefly, 50 μL of serum from each patient, starting from 1:10, in duplicate, in serial four-fold dilutions, were mixed in a flat-bottomed tissue culture microtiter plate (COSTAR, Washington DC, USA) with an equal volume of 100 TCID_50_ live SARS-CoV-2 viral strains: PV10734 (D614G, B.1), Delta (B.1.617.2), and Omicron (B.1.1.529, BA.1). The plates were incubated at 33 °C in 5% CO_2_ for 1 h, then 3 × 10^4^ VERO E6 cells (VERO C1008, Vero 76, clone 18 E6, Vero E6; ATCC CRL-1586) were added to each well and the plates were incubated at the same conditions for an additional 72 h. After 3 days of incubation, cells were stained with Gram’s crystal violet solution (Merck, Darmstadt, Germany) plus 5% formaldehyde 40% *m*/*v* (Carlo Erba S.p.A.) for 30 min. Wells were observed under a microscope to evaluate the degree of cytopathic effect compared with the control well. Blue staining of wells indicated vital cells, and hence the presence of nAbs. The nAb titer was the maximum dilution causing a 90% reduction in cytopathic effect. A positive anti-SARS-CoV-2 nAb titer was equal to or greater than 1:10. Positive and negative controls were included in all test runs.

### 2.5. Statistical Analysis

Continuous variables were reported as mean (±standard deviation) or median and interquartile range (IQR) as appropriate according to distribution, while categorical data were reported as numbers and percentages. Comparisons between groups were carried out with an independent *t*-test or Mann–Whitney U test for continuous variables and a chi-squared test or Fisher’s exact test for categorical variables, as appropriate. All statistical tests were two-sided, and associations were considered statistically significant when the values were below a nominal level of 0.05 (*p* < 0.05). Calculations were performed with IBM SPSS Statistics software version 24.

## 3. Results

Table 1 reports the characteristics of the 32 CCP donors. Their median age was 49 years (IQR 34.8–54.3 years). The male/female ratio was 9.7 (29 males and three females). We included all the CCP donations collected between October 2021 and March 2022. All CCP donors were vaccinated against COVID-19 (12 with mRNA vaccines, 10 with adenoviral vector vaccines, 10 with mixing vaccines) with a mean of 2.1 doses (±0.8) and had recovered from SARS-CoV-2 infection (B.1.617.2 VOC lineage in 19 cases and B.1.1.529 in 13 cases). The median days between last vaccine dose or COVID-19 recovery and CCP donation were 144 days (IQR 75.8–187.0 days) and 25 days (IQR 19.8–58.0 days), respectively. The mean anti-Delta nAb titer in the 19 B.1.617.2 infected donors was 322.6 (±239.4). The mean anti-Omicron nAb titer in the 13 B.1.1.529 infected donors was 232.3 (±200.9).

Table 2 reports characteristics and outcomes of the 21 consecutive COVID-19 patients receiving CCP. All patients received high-titer (nAb titer ≥ 160) CCP titered against the patient’s VOC lineage (B.1.617.2 in 14 cases and B.1.1.529 in seven cases). Their median age was 66 years (IQR 50–74 years). The male/female ratio was 4.3 (17 males and four females). Most of the patients were overweight (BMI ≥ 25: 14/21 (66.7%); median BMI 27.5, IQR 24.0–32.3). Approximately half (10/21, 47.6%) of the patients were not vaccinated against COVID-19. Among those vaccinated, one patient received one dose, five patients two doses, and the other five patients three doses. Ten patients were vaccinated with a COVID-19 mRNA vaccine, while only one patient with an adenoviral vector vaccine. The median days since last vaccine dose to COVID-19 were 98 days (IQR 34.0–148.5 days). All patients were hospitalized for severe COVID-19 between October 2021 and March 2022. Six patients were admitted in low-intensity departments while the other 15 patients in intermediate–high-intensity departments (12 patients with NIMV and three with IMV). The median PaO2/FiO2 before CCP transfusion was 122 (IQR 94.0–149.0). Notably, 85.7% (18/21) of patients had at least one comorbidity, while 66.7% (12/18) of them had three or more comorbidities. An underlying immunosuppressive status before SARS-CoV-2 infection was present in 47.6% of patients (10/21: 3 chronic lymphocytic leukemia, two non-Hodgkin’s lymphoma, one rheumatoid arthritis, one thymoma, one lung cancer, one prostatic cancer, one primary hemophagocytic lymphohistiocytosis). Of them, six patients were under active immunosuppressive therapy at the time of SARS-CoV-2 infection (one patient receiving methotrexate for rheumatoid arthritis, one patient receiving a chemotherapy protocol for thymoma, three patients receiving anti-CD20 monoclonal antibody therapy for B-cell lymphoproliferative disorders, one patient receiving ruxolitinib chemotherapy for primary hemophagocytic lymphohistiocytosis). In addition, three patients had severe infectious complications during hospitalization for COVID-19 (one fungal septicemia and two ventilator-associated bacterial pneumonia). Regarding the CCP-related data, the mean number of CCP units transfused per patient was 1.8 (±0.8). Overall, 37 CCP units were transfused to 21 COVID-19 patients, with a median nAb titer against the respective VOCs of 160 (IQR 160–640; mean nAb titer 363.2, ±205.3). Of them, nine patients (42.9%) received one unit; eight patients (38.1%) two units, and four patients (19%) three units. All patients had undetectable levels of anti-SARS-CoV-2 IgG antibodies before CCP transfusion. No adverse reactions to CCP transfusion were reported. The median time between symptom onset and CCP therapy was 9 days (IQR 6–11 days), while the median time between last CCP transfusion and viral clearance was 7 days (IQR 4–10 days). Regarding therapies for COVID-19 given concomitantly or prior to CCP therapy, all patients were under standard therapy, consisting of steroids ± heparin, during CCP infusion. Five patients (23.8%) had received remdesivir and two patients (9.5%) anti-SARS-CoV-2 mAbs. In addition to remdesivir and steroids, one patient (case no. 21) had received intravenous immunoglobulin and anakinra. A 28-day mortality rate of 14.3% (3/21) was reported. A subgroup analysis between alive and dead CCP-treated COVID-19 patients (Table 3) showed that patients who died were significantly older (78 years versus 60.5 years, *p* = 0.03), with a greater number of associated comorbidities (≥3 comorbidities: 100% versus 66.7%) and a more advanced disease (measured as median PaO2/FiO2: 86.0 versus 138.5, *p* = 0.03) than those surviving. In addition, CCP-treated patients who died received CCP units later (18 days versus 9 days, *p* = 0.02) than COVID-19 patients who recovered, although there was no difference in the mean number and nAb titer of the CCP units transfused between the two groups.

## 4. Discussion

In our hospital, CCP was extensively collected and utilized for hospitalized COVID-19 patients during the first two pandemic waves (February 2020–March 2021) with overall positive results [2,11]. During the third pandemic wave (April–August 2021), however, the use of CCP also declined in our hospital following the conflicting results from RCTs and the marketing of mAbs against SARS-CoV-2. A renewed interest towards CCP has been manifested by our clinicians since October 2021 due to the appearance of SARS-CoV-2 VOCs resistant to most of the mAb-based therapies available. Indeed, clinicians once again found themselves without an effective and specific anti-viral therapy for their patients, particularly for those not able to mount a sufficient anti-viral immune response. To meet the increasing CCP demands, we reorganized CCP collection using the best literature evidence to produce CCP with the highest nAb titer; thus, we recruited local regular blood donors that were both COVID-19 vaccinated and recently recovered from SARS-CoV-2 infection [25,26]. In addition, to give the maximum anti-viral effect, we transfused CCP units titered against patients’ VOCs and the Italian reference SARS-CoV-2 strain PV10734. This variant was first isolated in Pavia in 2020, and became the principal circulating variant completely replacing the Wuhan strain in Italy during the first wave [27]. The difference in terms of mutations resides in the substitution D614G in the spike protein. Nonetheless, the vaccine constructed with the Wuhan sequence maintains the same efficacy when tested with the PV10734 Italian reference strain (B.1, D614G) or the Wuhan original strain [22]. We modulated the CCP collection according to CCP requests from clinicians, keeping stocks at minimum to closely follow the continuous natural evolution of the virus and avoid later discarding of unmatched units. As a result, we have produced a potent vax-plasma specifically directed against the current local VOC. The high quality of CCP is also documented by the fact that the nAb content of CCP was found not to influence the outcome, as all CCP units transfused had high (>160) nAb titer against patients’ VOCs. All patients receiving CCP were seronegative for SARS-CoV-2; nearly 50% (10/21) were also immunosuppressed and the majority of them (6/10) were also under active immunosuppressive therapy at the time of COVID-19. Consequently, those patients vaccinated (nearly half were not COVID-19 vaccinated) were without an adequate antibody coverage due to their underlying immunosuppressive state or due to the long period of time elapsed between last vaccine dose and SARS-CoV-2 infection. The outcome of patients was positive in most cases with a mortality rate of 14.3%, which compares favorably with the historical mortality data on the COVID-19 general mortality rate (18.6%) in the City Hospital of Mantua [28]. As expected, younger patients with less severe disease and fewer comorbidities were those who benefited the most from hyperimmune plasma-based therapy, particularly when it was administered early from symptom onset. Notably, in five cases CCP was used as a last resort, after the failure of other anti-viral therapies (remdesivir and/or mAbs), and all patients survived. In particular, patients #3 and #15 were initially treated with remdesivir and mAbs (casirivimab + imdevimab) but, due to the worsening of their clinical conditions, they were subsequently treated (3 and 5 days post-mAb therapy, respectively), and effectively rescued, with CCP. A similar case has been reported in the literature [29]. Patient #21 is an exceptional case, that deserves a particular description. It is a 6-year-old child, unvaccinated, affected by a rare genetic disorder named primary hemophagocytic lymphohistiocytosis. This immunodeficient patient contracted a severe form of COVID-19 that required admission to the Intensive Care Unit of the Santobono Pausilipon Hospital of Naples and invasive mechanical ventilation. Several anti-viral and immunomodulatory agents were utilized, including steroids, intravenous immunoglobulin, remdesivir, and anakinra. The patient had been previously transfused with a CCP unit collected locally (Campania region), but one year before and not titered against the Delta variant (the patient’s VOC lineage). All these therapies had a scant effect on the disease course. As a last resort, considering the patient’s persisting molecular positivity for SARS-CoV-2, clinicians decided to request two ABO-matched CCP units, with high nAb titer against the Delta variant, at the Blood Bank of Mantua. The two units were transfused 7 days apart (the first CCP unit was transfused 15 days from symptom onset). Following the second CCP transfusion, there was a slow but progressive clinical improvement and 21 days later the patient persistently cleared the virus and thus was transferred to the hemato-oncological unit to continue chemotherapies. This case is, however, also interesting for another aspect. High-nAb CCP, indeed, was transfused late in the disease course but it was equally effective, suggesting that in such immunosuppressed patients the natural history of COVID-19 is quite different from that of immunocompetent individuals and that CCP timing is not a primary issue. In other words, in immunodeficient patients high-titer, VOC-matched CCP could also be effective if given later than the usual 3–5 days from symptom onset. 

Our study has several limitations, including the retrospective and heterogeneous nature of the cohort, the small sample size, and the lack of a control group. The rarity of such underlying comorbidities makes randomized controlled studies feasible only in large multicenter trials. Nevertheless, our single-institution experience comes with the strength of homogeneous validation criteria for CCP.

In conclusion, this real-life experience on the compassionate use of CCP in selected categories of patients without antibodies against SARS-CoV-2 hospitalized for severe COVID-19 was favorable in terms of safety and efficacy and once again supports its use, including during Delta and Omicron pandemic waves.

## Figures and Tables

**Table 1 viruses-14-01443-t001:** Characteristics of the COVID-19 convalescent plasma donors.

CCP Donorno.	Sex	Age (Years)	CCP Donation (m/d/y)	COVID-19 Vaccine			Anti-SARS-CoV-2 nAb Titer			Days between CCP Donation and		VOC Lineage
				Type	Doses	Last Dose (Month/Day/Year)	Anti-Wild Type	Anti- B.1.617.2	Anti- B.1.1.529	Last Vaccine Dose	COVID-19 Recovery	
1	M	53	10/06/2021	AZ	2	06/09/2021	640	320	10	119	13	B.1.617.2
2	M	56	10/20/2021	AZ	1	05/15/2021	640	80	<10	158	89	B.1.617.2
3	M	27	10/27/2021	AZ	2	05/15/2021	640	160	10	165	141	B.1.617.2
4	F	35	10/27/2021	Pfizer	1	06/26/2021	160	40	<10	123	70	B.1.617.2
5	M	46	10/27/2021	J&J	1	06/06/2021	640	160	10	143	19	B.1.617.2
6	M	20	11/10/2021	Pfizer	1	06/20/2021	320	160	10	143	82	B.1.617.2
7	F	52	11/10/2021	Pfizer	2	03/10/2021	640	320	40	245	71	B.1.617.2
8	M	35	11/17/2021	Pfizer	1	02/02/2021	640	640	40	288	54	B.1.617.2
9	M	53	11/24/2021	AZ	1	06/07/2021	20 ^1^	10 ^1^	10 ^1^	170	91	B.1.617.2
10	M	58	12/01/2021	Pfizer	2	07/06/2021	160	80	<10	148	20	B.1.617.2
11	M	46	12/10/2021	Pfizer	2	07/18/2021	320	160	10	145	21	B.1.617.2
12	M	51	12/10/2021	J&J	1	06/07/2021	640	320	20	186	35	B.1.617.2
13	M	28	12/15/2021	AZ	2	05/31/2021	320	640	40	198	23	B.1.617.2
14	M	53	12/16/2021	AZ	2	06/09/2021	640	160	10	190	84	B.1.617.2
15	M	26	12/22/2021	J&J	1	06/30/2021	640	640	40	175	27	B.1.617.2
16	F	47	12/27/2021	Pfizer	2	06/10/2021	NP	320	40	200	14	B.1.617.2
17	M	51	12/27/2021	AZ	2	06/09/2021	NP	640	160	201	26	B.1.617.2
19	M	26	01/05/2022	Pfizer	2	08/05/2021	NP	640	10	153	21	B.1.617.2
19	M	59	01/05/2022	Pfizer	2	06/26/2021	NP	640	<10	193	149	B.1.617.2
20	M	40	02/08/2022	Pfizer	2	10/29/2021	NP	640	160	105	22	B.1.1.529
21	M	57	02/08/2022	Pfizer	3	10/26/2021	NP	640	320	102	18	B.1.1.529
22	M	57	02/11/2022	Pfizer (2)/AZ (1)	3	11/29/2021	NP	640	640	74	36	B.1.1.529
23	M	56	02/16/2022	Pfizer (2)/Moderna (1)	3	12/18/2021	NP	640	160	60	21	B.1.1.529
24	M	55	02/17/2022	Pfizer (2)/AZ (1)	3	12/03/2021	NP	320	40	76	22	B.1.1.529
25	M	56	02/18/2022	Pfizer (2)/AZ (1)	3	01/20/2022	NP	160	80	29	18	B.1.1.529
26	M	34	02/21/2022	Pfizer (2)/Moderna (1)	3	01/12/2022	NP	640	160	40	19	B.1.1.529
27	M	51	02/24/2022	J&J (1)/Moderna (2)	3	12/23/2021	NP	320	160	63	32	B.1.1.529
28	M	40	03/11/2022	Pfizer	3	07/17/2021	NP	160	20	237	24	B.1.1.529
29	M	24	03/15/2022	Pfizer (2)/Moderna (1)	3	01/13/2022	NP	160	160	61	45	B.1.1.529
30	M	34	03/22/2022	Pfizer (2)/Moderna (1)	3	12/16/2021	NP	640	160	96	17	B.1.1.529
31	M	54	03/23/2022	Pfizer (2)/Moderna (1)	3	01/07/2022	NP	640	320	75	13	B.1.1.529
32	M	38	03/25/2022	Pfizer (2)/Moderna (1)	3	01/21/2022	NP	640	640	63	52	B.1.1.529

Abbreviations: CCP, COVID-19 convalescent plasma; AZ, AstraZeneca; J&J, Janssen-Johnson & Johnson; NP, not performed; VOC, variant of concern. ^1^ Unit from donor 9 was not qualified as CCP due to insufficient neutralizing antibody activity (<160 to all VOCs).

**Table 2 viruses-14-01443-t002:** Characteristics and outcome of COVID-19 patients receiving CCP.

Pt.no.	Age (Years)	Sex	BMI	COVID-19 Vaccine/Doses	Days Since LastVaccine Dose to COVID-19	Hospital Admission (m/d/y)	VOC Lineage	COVID-19 Severity/Hospital Department	Comorbidities/Complications	PaO2/FiO2 before CCP Transfusion	Anti-SARS-CoV-2 IgG ^1^	Number of CCP Units Transfused/nAb Titer ^2^	Days between Symptom Onset and CCP Transfusion	Days between Last CCP Transfusion and VC	Previous/Concomitant Therapies	Outcome ^3^
1	75	M	24	No	-	10/29/2021	B.1.617.2	Severe/RU	AF, hypertension	140	Negative	3/320	14	11	ST, HEP	Alive
2	60	M	34	Pfizer/2	160	11/16/2021	B.1.617.2	Severe/RU	Hypertension, diabetes, CT for thymoma	156	Negative	3/160–320	10	10	ST, HEP, REM	Alive
3	52	M	33	No	-	11/18/2021	B.1.617.2	Severe/RICU	CRU, VABP, tracheostomy	115	Negative	3/160	6	18	ST, HEP, REM, mAbs ^4^	Alive
4	28	F	31	No	-	11/17/2021	B.1.617.2	Severe/RICU	Pituitary adenoma	95	Negative	3/160–640	7	3	ST, HEP	Alive
5	50	F	29	No	-	09/21/2021	B.1.617.2	Severe/RU	Hypertension, tracheostomy	94	Negative	1/160	9	7	ST, HEP	Alive
6	72	M	23	Pfizer/3	98	01/08/2022	B.1.617.2	Severe/RU	NHL	118	Negative	2/320	11	NA	RTX, ST, HEP	Alive
7	72	M	27	Moderna/3	26	02/26/2022	B.1.1.529	Severe/RU	Diabetes, hypertension	149	Negative	1/640	6	7	ST, HEP	Alive
8	78	M	35	AZ/2	142	11/17/2021	B.1.617.2	Severe/RICU	Diabetes, hypertension, PC	88	Negative	2/160	18	NA	ST, HEP	Dead
9	77	M	26	No	-	12/21/2021	B.1.617.2	Severe/RICU	CAD, hypertension, dyslipidemia, LC	72	Negative	2/640	19	NA	ST, HEP	Dead
10	72	M	28	No	-	12/28/2021	B.1.617.2	Severe/RICU	Diabetes, MI, hypertension, FS	144	Negative	2/160–640	9	4	ST, HEP	Alive
11	66	M	37	Pfizer/2	176	12/30/2021	B.1.617.2	Severe/RICU	OSAS, diabetes, dyslipidemia, CAD	91	Negative	1/640	11	10	ST, HEP	Alive
12	44	M	32	No	-	01/29/2022	B.1.1.529	Severe/ICU	Hypertension, VABP	122	Negative	1/160	5	9	ST, HEP	Alive
13	81	M	28	Moderna/1	2	12/23/2021	B.1.617.2	Severe/IMU	Hypertension, CPOA, SCI	86	Negative	1/320	6	NA	ST. HEP	Dead
14	49	M	33	No	-	12/31/2021	B.1.617.2	Severe/IMU	None	175	Negative	1/160	10	7	ST, HEP	Alive
15	72	M	23	Pfizer/3	42	01/12/2022	B.1.617.2	Severe/IMU	CLL	144	Negative	2/640	9	NA	ST, HEP, REM, mAbs ^4^	Alive
16	61	F	21	No	-	01/24/2022	B.1.1.529	Severe/IMU	None	117	Negative	1/160	6	4	ST, HEP, REM	Alive
17	54	M	24	Pfizer/2	84	01/26/2022	B.1.1.529	Severe/IMU	None	259	Negative	1/160	8	8	ST, HEP	Alive
18	50	M	26	Moderna/2	22	01/22/2022	B.1.1.529	Severe/IMU	RA, ^5^ hypertension	244	Negative	1/160	11	5	ST, HEP, MTX	Alive
19	74	F	22	Moderna/3	153	03/03/2022	B.1.1.529	Severe/RICU	NHL	81	Negative	2/320	1	NA	ST, HEP, RTX	Alive
20	76	M	25	Pfizer/3	144	03/16/2022	B.1.1.529	Severe/RICU	CLL, CAD, hypertension	147	Negative	2/640	3	3	ST, HEP, RTX	Alive
21	6	M	14	No	-	01/14/2022	B.1.617.2	Severe/ICU	PHLH	230	Negative	2/320–640	15	21	ST, REM, IVIG, ANA, RUX	Alive

Legend: M, male; F, female; NA, not available; AF, atrial fibrillation; RU, respiratory unit; RICU, respiratory intensive care unit; IMU, internal medicine unit; AZ, AstraZeneca; BMI, body mass index; ST, steroids, HEP, heparin; mAbs, anti-COVID-19 monoclonal antibodies; REM, remdesivir; CT, chemotherapy; CRU, colitis rectal ulcerous; VABP, ventilator-associated bacterial pneumonia; non-Hodgkin lymphoma; RTX, rituximab; PC, prostatic cancer; LC, lung cancer; FS, fungal septicemia; OSAS, obstructive sleep apnea syndrome; MI, myocardial ischemia; CAD, coronary artery disease; ICU, intensive care unit; CPOA, chronic peripheral obstructive arteriopathy, SCI, severe cognitive impairment; CLL, chronic lymphocytic leukemia; RA, rheumatoid arthritis; VC, viral clearance; PHLH, primary hemophagocytic lymphohistiocytosis; IVIG, intravenous aspecific immunoglobulin; ANA, anakinra; RUX, ruxolitinib; MTX, methotrexate. ^1^ Patients #3 and #5 were seronegative before mAb infusion. ^2^ nAb titered against VOC. ^3^ Twenty-eight days after hospitalization. ^4^ Casirivimab + imdevimab. ^5^ Under chronic steroid therapy.

**Table 3 viruses-14-01443-t003:** Characteristics of CCP-treated COVID-19 patients, dead and alive.

Parameters	Alive n = 18 (85.7%)	Dead n = 3 (14.3%)	*p*
Median age, years (IQR)	60.5 (50.0–72.0)	78 (77.5–79.5)	0.03
Sex (males/females), number	15/3	3/0	-
Median BMI (kg/m^2^) (IQR)	27.0 (24.0–32.0)	28.0 (27.0–31.5)	NS
Comorbidities, n (%) <3 ≥3	6 (33.3) 12 (66.7)	0 (0) 3 (100)	-
PaO^2^/FiO^2^, median (IQR)	138.5 (115.5–154.3)	86.0 (79.0–87.0)	0.03
Days between symptom onset and CCP therapy, median (IQR)	9 (6.0–10.8)	18 (12.0–18.5)	0.02
CCP units transfused, mean (±SD)	1.8 (±0.8)	1.7 (±0.6)	NS
CCP neutralizing titer, mean (±SD)	360.0 (±203.2)	384.0 (±242.6)	NS

Abbreviations: NS, not significant; CCP, COVID-19 convalescent plasma; BMI, body mass index; SD, standard deviation.

## Data Availability

The data presented in this study are available on request from the corresponding author.

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
