# Peer review of "Variant of Concern-Matched COVID-19 Convalescent Plasma Usage in Seronegative Hospitalized Patients"

_viruses, 2022, doi:10.3390/v14071443_

Round 1

Reviewer 1 Report

I listed the major and minor comments below.

 Major Comments:

1)      The author mentioned the

2)      The authors didn’t describe how they determined the VOC lineage for both donors and recipients for the convalescent plasma transfusion;

3)      The key lab experiment assay -viral neutralization test should be described in more detail in the Materials and Methods (MM) section. The reader is interested in knowing which virus system they used for this assay, live virus or pseudovirus? How did they determine the titers?

4)      The each COVID-19 vaccine from different vendors should be briefly described in the MM section;

5)      The wide-type the authors referred to is Wuan Strain? If not, please detail the difference between the WT author referred to and Wuhan strain, which is the most of current vaccine used to encode the vaccine immunogens.

6)      In the table 1, some donors received 3 dose, please detail each dose vaccine from which vendor.

7)      In the table 1, please use the same terminology (B.1.617.2 and B.1.1.529) to describe the delta, omicron?

8)      Please explain why neutralizing antibody assay against wide-type was not performed for some donors?

9)      The table 2, can the authors separately list the age and sex? Can the authors further classify the outcome for the alive?

10)  The table 2, what does it mean by using “NA”, “-”,”NO”?

11)  In the Discussion section, the authors mentioned two pandemic waves but only provided one. Can the authors provide another one?

 Minor Comments:

Please verify that the plasma or serum were collected from the donors. In the line 154, the author mentioned the serum.

Author Response

Major Comments:

2)      The authors didn’t describe how they determined the VOC lineage for both donors and recipients for the convalescent plasma transfusion;

Answer: VOC identification was performed using an in house primers set to identify variants defying mutations as reported in Gaiarsa S. et al. 2022.

3)      The key lab experiment assay -viral neutralization test should be described in more detail in the Materials and Methods (MM) section. The reader is interested in knowing which virus system they used for this assay, live virus or pseudovirus? How did they determine the titers?

Answer: The viral neutralization assay description in the Material and methods section was revised (from lane 139 to 162) and implemented as requested by the reviewer. For all test we used live virus, thus we changed the text as follows: “…100 TCID50 live SARS-CoV-2 viral strains…”. We also added a brief description of the viral stock titer determination from lane 142 to 145.

4)      The each COVID-19 vaccine from different vendors should be briefly described in the MM section;

Answer: done.

5)      The wide-type the authors referred to is Wuan Strain? If not, please detail the difference between the WT author referred to and Wuhan strain, which is the most of current vaccine used to encode the vaccine immunogens.

Answer: The wild type strain we are referring to is the Italian strain, first isolated in Pavia in 2020, and which became the principal circulating strain completely replacing the Wuhan strain in Italy during the first wave [Alteri C et al., 2020]. The difference in terms of mutations resides in the substitution D614G in the Spike protein.  Noteworthy, the efficacy of the vaccine constructed with the Wuhan sequence is maintained at the same level when tested with the PV10734 Italian reference strain (B.1, D614G) [Sammartino et al., 2022]. We added this information in the text as well line 236 to 241.

  • Reed, L.J.; Muench, H. A simple method of estimating fifty percent endpoints. J. Hygiene 1938, 27, 493–497.
  • Alteri C, Cento V, Piralla A, Costabile V, Tallarita M, Colagrossi L, Renica S, Giardina F, Novazzi F, Gaiarsa S, Matarazzo E, Antonello M, Vismara C, Fumagalli R, Epis OM, Puoti M, Perno CF, Baldanti F. Genomic epidemiology of SARS-CoV-2 reveals multiple lineages and early spread of SARS-CoV-2 infections in Lombardy, Italy. Nat Commun. 2021 Jan 19;12(1):434. doi: 10.1038/s41467-020-20688-x. PMID: 33469026; PMCID: PMC7815831.
  • Sammartino JC, Cassaniti I, Ferrari A, Giardina F, Ferrari G, Zavaglio F, Paolucci S, Lilleri D, Piralla A, Baldanti F, Percivalle E. Evaluation of the Neutralizing Antibodies Response against 14 SARS-CoV-2 Variants in BNT162b2 Vaccinated Naïve and COVID-19 Positive Healthcare Workers from a Northern Italian Hospital. Vaccines (Basel). 2022 Apr 29;10(5):703. doi: 10.3390/vaccines10050703. PMID: 35632457; PMCID: PMC9145000.
  • Gaiarsa S, Giardina F, Batisti Biffignandi G, Ferrari G, Piazza A, Tallarita M, Novazzi F, Bandi C, Paolucci S, Rovida F, Campanini G, Piralla A, Baldanti F. Comparative analysis of SARS-CoV-2 quasispecies in the upper and lower respiratory tract shows an ongoing evolution in the spike cleavage site. Virus Res. 2022 Jul 2;315:198786. doi: 10.1016/j.virusres.2022.198786. Epub 2022 Apr 14. PMID: 35429618; PMCID: PMC9008095.

6)      In the table 1, some donors received 3 dose, please detail each dose vaccine from which vendor.

Answer: done.

7)      In the table 1, please use the same terminology (B.1.617.2 and B.1.1.529) to describe the delta, omicron?

Answer: done.

8)      Please explain why neutralizing antibody assay against wide-type was not performed for some donors?

Answer: It was not performed for some donors because this strain was no longer present in Italy.

9)      The table 2, can the authors separately list the age and sex? Can the authors further classify the outcome for the alive?

Answer: done.

10)  The table 2, what does it mean by using “NA”, “-”,”NO”?

Answer: I have clarified these terms.

11)  In the Discussion section, the authors mentioned two pandemic waves but only provided one. Can the authors provide another one?

Answer: This study was performed during the last pandemic wave. The results of the first two pandemic waves have already been published (reference no. 11).

 Minor Comments:

 Please verify that the plasma or serum were collected from the donors. In the line 154, the author mentioned the serum.

Answer: done.

Reviewer 2 Report

Franchini et al. describe a series of 21 patients treated with plasma from donors recovered from and vaccinated against COVID-19 (CCP). The paper highlights the potential benefit of high titer VOC matched plasma in patients with severe COVID and negative for anti-SARS-CoV-2 antibodies due to some degree of immunosupression. The paper is well-written and the number of patients included (21) is sufficient to support further studies on the benefits of CCP in this patient population. The authors should consider to address the following points:

-          Table 1. As for the authors explanations in the text, are the plasma from donors 9,2, 9, 24, 25 and 28 have not qualify as CCP due to insufficient nAb activity (<160 to the matched VOC)? If so, state it clearly in the table 1.

-          Table 2. It is not clear to this reviewer which plasma unit was transfused to patient 7, a male infected with Omicron transfused early February 2022 with a CCP unit with 640 nAB titer (admission date Jan 26, transfusion 6 days after symptoms onset, presumably before hospital admission?). This reviewer is not able to find the corresponding unit in table 1. Donors 20 and 21 have nAb of 160 and 320, respectively. A column should be added in table 2 to link the CCP plasma from donors in table 1 to the respective recipient.

-          The authors should clarify how they determined the VOC in the plasma donors, or they assumed the VOC by the circulating variant in the time of donors’ COVID infection.   

-          In the discussion section, authors should be cautious to stress the relevance of VOC-matched CCP from vaccinated donors. It is probable that this association is relevant for Omicron infected patients, but for Delta the nAb come mainly from the vaccine boosters, independently of the variant infecting the donor. The design of the study does not permit to compare the clinical efficacy of VOC-matched plasma against the non-VOC-matched plasma.   

-          Discussion section, page 12, line29: The authors should further comment on the “historical mortality data of the City of Mantua”. The 14.3% mortality rate in the paper comperes favorably, but the exact mortality rate on general population should be mention alongside, rather to refer the reader to the original reference.

As minor revision points:

-          Abstract: RCT is mentioned for the first time in the paper. Please change to “randomized controlled trials”. Accordingly, substitute non-randomized controlled trials (Introduction, page 2 line 49) for non-RCTs.

-          Methods section, 2.3 COVID convalescent plasma. Authors should clarify whether the plasma is frozen util transfused, and if any pathogen reduction technologies has been used. If so, a clarification on the potential loss of nAb activity should be discussed in the Discussion section.

-          Results section, page 5 line 210: “Of them, 6 patients…” do the authors mean 6 patients out of the 10 immunosupressed patients?

-          In table 1, the authors should consider to include the value of the Diasorin Ab quantitation, in order to help comparison with other studies using commercial antibody quantitation.

-          In table 2, Viral clearance column: What does NA stand for? It is not included in the table legend. Does the slash mean not recorded?   

-          Table 3: Comorbidities, it is not necessary to repeat the denominator (Alive N=18, Dead N=3; already mentioned in the column heading). Substitute “Died” for “Dead” in the column heading. The P value is missing for the sex and comorbidities files.

Author Response

Franchini et al. describe a series of 21 patients treated with plasma from donors recovered from and vaccinated against COVID-19 (CCP). The paper highlights the potential benefit of high titer VOC matched plasma in patients with severe COVID and negative for anti-SARS-CoV-2 antibodies due to some degree of immunosupression. The paper is well-written and the number of patients included (21) is sufficient to support further studies on the benefits of CCP in this patient population. The authors should consider to address the following points:

-          Table 1. As for the authors explanations in the text, are the plasma from donors 9,2, 9, 24, 25 and 28 have not qualify as CCP due to insufficient nAb activity (<160 to the matched VOC)? If so, state it clearly in the table 1.

Answer: done.

-          Table 2. It is not clear to this reviewer which plasma unit was transfused to patient 7, a male infected with Omicron transfused early February 2022 with a CCP unit with 640 nAB titer (admission date Jan 26, transfusion 6 days after symptoms onset, presumably before hospital admission?). This reviewer is not able to find the corresponding unit in table 1. Donors 20 and 21 have nAb of 160 and 320, respectively. A column should be added in table 2 to link the CCP plasma from donors in table 1 to the respective recipient.

Answer: I have corrected the date. I apologize for the mistake. I think that this column would be not informative for the readers.

-          The authors should clarify how they determined the VOC in the plasma donors, or they assumed the VOC by the circulating variant in the time of donors’ COVID infection.

Answer. Done.   

-          In the discussion section, authors should be cautious to stress the relevance of VOC-matched CCP from vaccinated donors. It is probable that this association is relevant for Omicron infected patients, but for Delta the nAb come mainly from the vaccine boosters, independently of the variant infecting the donor. The design of the study does not permit to compare the clinical efficacy of VOC-matched plasma against the non-VOC-matched plasma. 

Answer. I have added the limit of the study of the lack of a control group.  

-          Discussion section, page 12, line29: The authors should further comment on the “historical mortality data of the City of Mantua”. The 14.3% mortality rate in the paper compares favorably, but the exact mortality rate on general population should be mention alongside, rather to refer the reader to the original reference.

Answer: done.

As minor revision points:

-          Abstract: RCT is mentioned for the first time in the paper. Please change to “randomized controlled trials”. Accordingly, substitute non-randomized controlled trials (Introduction, page 2 line 49) for non-RCTs.

Answer: Done.

-          Methods section, 2.3 COVID convalescent plasma. Authors should clarify whether the plasma is frozen util transfused, and if any pathogen reduction technologies has been used. If so, a clarification on the potential loss of nAb activity should be discussed in the Discussion section.

Answer: CCP was frozen (this information was added). No pathogen reduction technology was used.

-          Results section, page 5 line 210: “Of them, 6 patients…” do the authors mean 6 patients out of the 10 immunosupressed patients?

Answer: Yes.

-          In table 1, the authors should consider to include the value of the Diasorin Ab quantitation, in order to help comparison with other studies using commercial antibody quantitation.

Answer: This test was not performed in CCP donors but only in patients. I have added this information.

-          In table 2, Viral clearance column: What does NA stand for? It is not included in the table legend. Does the slash mean not recorded? 

Answer: it was clarified.  

-          Table 3: Comorbidities, it is not necessary to repeat the denominator (Alive N=18, Dead N=3; already mentioned in the column heading). Substitute “Died” for “Dead” in the column heading. The P value is missing for the sex and comorbidities files.

Answer: done.

Reviewer 3 Report

This is a well-written paper which describes the therapeutic potential of personalized, Variant of Concern matched and antiviral Neutralizing Antibody tittered convalescent plasma in hospitalized patients with COVID-19 who, at the time of plasma dosing, were seronegative for anti-spike SARS-CoV-2 antibodies and were clinically deteriorating despite having received, or while receiving, standard of care treatment with steroids, heparin, Remdesivir, and Monoclonal Antibodies in various combinations.

I would thoroughly recommend its publication because, at present, the therapeutic armamentarium against SARS-CoV-2 is severely crippled by the successive circulation of SARS-CoV-2 Variants of Concern that exhibit an ever-growing immune evasion and, consequently, resistance to the commercially available Monoclonal Antibodies and to vaccination-induced humoral immunity.

However, I feel the Authors should devote some space to the acknowledgement of the limitations of their work.

In particular, except for the time to viral clearance from the nasopharynx, no information is provided on other viral markers such as the viral load and live virus shedding in this anatomical district, at baseline and following convalescent plasma dosing. No data on SARS-CoV-2 RNAemia are given as well.

No data are presented regarding anti-Spike and anti-Nucleocapsid serologies following convalescent plasma dosing, and the same is true for viral neutralization titers.

In addition, the is no mention of anti-SARS-CoV-2 cell mediated immunity before and after administration of convalescent plasma.

Finally, the number of patients is limited, and this should be emphasized.

Author Response

This is a well-written paper which describes the therapeutic potential of personalized, Variant of Concern matched and antiviral Neutralizing Antibody tittered convalescent plasma in hospitalized patients with COVID-19 who, at the time of plasma dosing, were seronegative for anti-spike SARS-CoV-2 antibodies and were clinically deteriorating despite having received, or while receiving, standard of care treatment with steroids, heparin, Remdesivir, and Monoclonal Antibodies in various combinations.

I would thoroughly recommend its publication because, at present, the therapeutic armamentarium against SARS-CoV-2 is severely crippled by the successive circulation of SARS-CoV-2 Variants of Concern that exhibit an ever-growing immune evasion and, consequently, resistance to the commercially available Monoclonal Antibodies and to vaccination-induced humoral immunity.

However, I feel the Authors should devote some space to the acknowledgement of the limitations of their work.

In particular, except for the time to viral clearance from the nasopharynx, no information is provided on other viral markers such as the viral load and live virus shedding in this anatomical district, at baseline and following convalescent plasma dosing. No data on SARS-CoV-2 RNAemia are given as well.

No data are presented regarding anti-Spike and anti-Nucleocapsid serologies following convalescent plasma dosing, and the same is true for viral neutralization titers.

In addition, the is no mention of anti-SARS-CoV-2 cell mediated immunity before and after administration of convalescent plasma.

Finally, the number of patients is limited, and this should be emphasized.

Answer: We have strengthened the limitations of our study in the Discussion.

Reviewer 4 Report

This elegant article describes clearly and in deep the positive experience of Mantua hospital in Italy in transfusion of COVID-19 convalescent plasma enriched in neutralizing antibodies against the two recent variants of concern , namely Delta and Omicron  from October 2021 to March 2022 in  21 immunocompromised patients hospitalized for severe COVID disease. Taking into consideration that the determination of specific neutralizing antibodies titer of these convalescent plasma are of paramount importance, I advise the authors to describe it in deep. It is obvious that this article is typically in the scope of the special issue of Viruses Journal on the "State of the Art -SARS-CoV2 Research in Italy.

Author Response

This elegant article describes clearly and in deep the positive experience of Mantua hospital in Italy in transfusion of COVID-19 convalescent plasma enriched in neutralizing antibodies against the two recent variants of concern , namely Delta and Omicron  from October 2021 to March 2022 in  21 immunocompromised patients hospitalized for severe COVID disease. Taking into consideration that the determination of specific neutralizing antibodies titer of these convalescent plasma are of paramount importance, I advise the authors to describe it in deep. It is obvious that this article is typically in the scope of the special issue of Viruses Journal on the "State of the Art -SARS-CoV2 Research in Italy.

Answer: done.